# Problem-solving training: assessing the feasibility and acceptability of delivering and evaluating a problem-solving training model for front-line prison staff and prisoners who self-harm

Amanda Perry [1], Mitchell Glenn Waterman,[2] Allan House [3], Alexandra Wright-Hughes,[4] Joanne Greenhalgh,[5] Amanda Farrin,[4] Gerry Richardson,[6] Ann Kathryn Hopton,[1] Nat Wright[7]

For numbered affiliations see end of article.

**Correspondence to**
Dr Amanda Perry;
amanda.perry@york.ac.uk

## ABSTRACT

**Objectives** Problem-solving skills training is adaptable, inexpensive and simple to deliver. However, its application with prisoners who self-harm is unknown. The study assessed the feasibility and acceptability of a problem-solving training (PST) intervention for prison staff and prisoners who self-harm, to inform the design of a large-scale study.

**Design and setting** A mixed-methods design used routinely collected data, individual outcome measures, an economic protocol and qualitative interviews at four prisons in Yorkshire and Humber, UK.

**Participants** (i) Front-line prison staff, (ii) male and female prisoners with an episode of self-harm in the previous 2 weeks.

**Intervention** The intervention comprised a 1 hour staff training session and a 30 min prisoner session using adapted workbooks and case studies.

**Outcomes** We assessed the study processes—coverage of training; recruitment and retention rates and adequacy of intervention delivery—and available data (completeness of outcome data, integrity of routinely collected data and access to the National Health Service (NHS) resource information). Prisoner outcomes assessed incidence of self-harm, quality of life and depression at baseline and at follow-up. Qualitative findings are presented elsewhere.

**Results** Recruitment was higher than anticipated for staff n=280, but lower for prisoners, n=48. Retention was good with 43/48 (89%) prisoners completing the intervention, at follow-up we collected individual outcome data for 34/48 (71%) of prisoners. Access to routinely collected data was inconsistent. Prisoners were frequent users of NHS healthcare. The additional cost of training and intervention delivery was deemed minimal in comparison to 'treatment as usual'. Outcome measures of self-harm, quality of life and depression were found to be acceptable.

**Conclusions** The intervention proved feasible to adapt. Staff training was delivered but on the whole it was not deemed feasible for staff to deliver the intervention. A

### Strengths and limitations of this study

► Prison staff and prisoners were involved in the development of our questionnaires, the intervention adaptation and production of the workbooks.
► The feasibility study was conducted across four prison sites including male and female prisoners.
► Outcome data were collected via a variety of different sources demonstrating variability and differences in data collection procedures.
► It was not deemed feasible for staff to deliver the intervention.

large-scale study is warranted, but modifications to the implementation of the intervention are required.

## INTRODUCTION

Problem-solving skills training delivered in a systematic manner provide a non-specialist intervention that is accessible to anyone following brief training. Deficits in problem-solving skills are often found in people who self-harm and can result in reliance on others, leading to a passive as opposed to an active problem-solving approach.[1–3] Problem-solving skills have been used in a variety of different contexts and most recently are promoted by WHO as 'Problem Management Plus'.[4] They refer to their scheme as a simplified, scalable intervention, in that their delivery requires a less intensive level of specialist human resource use.[5] Trials of problem-solving skills in the *community* demonstrate that teaching people to use brief problem-solving skills can reduce repetition of self-harm behaviour.[6–8]

In prison, despite growing numbers of those who self-harm there is a lack of psychological support for prisoners and a recognised need to provide adequate staff training (National Institute for Health and Care Excellence Guidance CG133: https://www.nice.org.uk/guidance/cg133/chapter/2-Research-recommendations). Evaluations of trials in prisons have explored alternative therapy models for those who self-harm (eg, cognitive behaviour therapy and interpersonal psychotherapy), but such interventions require the use of extensive resources, large numbers of therapy sessions and qualified clinical therapists, making them inaccessible for prisoners who might only be incarcerated for short periods of time.[9 10]

Use of a brief problem-solving training (PST) intervention offers one solution to this problem. It has the advantages of being deliverable by any member of staff making it an attractive, inexpensive opportunity to reduce repeat self-harm. However, it is unclear whether the training is acceptable, or whether it can be implemented by staff in this setting. We therefore assessed the feasibility and acceptability of adapting an existing PST for front-line prison staff with the intention that they would deliver the intervention to prisoners who self-harm. This article reports on the acceptability and feasibility of the training, and the implementation of the intervention. Detailed methods on the qualitative findings are mentioned elsewhere.[11]

## MATERIALS AND METHODS
### Study design and setting
The study used a mixed-methods design—including quantitative collection of routine data, individual outcome measures and economic resource data and information from staff to identify how much time was spent on 'usual care'.

The study took place in four prisons in Yorkshire and Humber between September 2014 and May 2017. The study sites included two male adult local prisons where the majority of prisoners are awaiting sentence (housing up to 1212 and 1052 prisoners, prisons A and B), one female prison (housing up to 416: prison C) and one resettlement prison where sentenced prisoners are housed prior to transfer (housing up to 825: prison D). We report on our intervention using the Template for Intervention Description and Replication checklist.[12]

### Patient and public involvement
Our research questions and outcome measures were not informed by prisoner preferences and prisoners were not involved in the recruitment to the study. Prisoners did contribute significantly to the format and adaptation of the training materials. The training materials were printed from within the prison by prisoners. The results were disseminated using an A4 summary sheet, which was sent to prisoners and prison staff.

### The intervention
The PST intervention that we adapted for use in our study was originally devised in New Zealand for patients who self-harm in the community.[13] The theory behind social problem-solving is well established and often forms part of more extensive cognitive behaviour therapy sessions.[14 15] The seven-step model includes 'getting the right attitude' (step one), reflection and recognising triggers (step two), defining a clear problem (step three), brain storming solutions (step four), decision making (step five), making a plan (step six) and reviewing progress (step seven). Problem-solving skills are an approach that encourages an individual to address their problems in a proactive manner using the systematic seven-step process.

### The adaptation of the training and intervention materials
The training was adapted using focus groups. They were used to ensure (i) the appropriateness and context of the case materials and (ii) to promote discussion with staff and prisoners about their views on how the study might work. The refinement process involved a series of structured discussions facilitated by the research team to inform literacy levels in the population and scenario situations that could be used in training as examples of people that staff and prisoners could recognise and/or deal with on a regular basis.

### Staff training and recruitment
Staff were recruited with the help of prison liaison staff who assisted with room bookings, shift management and allocation of individuals to attend the training course. Using estimates provided by the prison about: the number, and type of staff employed by the prison, we estimated a feasible recruitment goal of 125 trained staff across the four sites in our 12-month training period.

Staff received a 1 hour training session between March 2015 and August 2016. Training was delivered by the research team in a flexible manner (eg, during induction or on a lunchtime). Eligible prison staff included anyone with responsibility for prisoners who were at risk of self-harm and who were monitored under the prison system (Assessment Care in Custody Teamwork (ACCT)[16]). Invited staff groups included management, probation, teaching, prison officers, chaplaincy, psychologists, specialist suicide prevention assessors and nursing staff. All staff receiving the training gave full informed consent.

### Recruitment and implementation of the intervention with prisoners who self-harmed
Recruitment of prisoners occurred at prison sites A, B and D. In site C, access to the prison site was limited. Our feasible recruitment goal of 120 were based on access to three sites and monthly prison information on the numbers of those 'at risk'.

Prisoners were identified via the ACCT register and approached by a member of the research team or prison staff. Eligible prisoners were (1) aged >16 years or over and (2) with an episode of self-harm or attempted suicide in the previous 2 weeks. Prisoners were excluded if (i) an ACCT was opened for reasons other than actual self-harm or attempted suicidal behaviour, (ii) they were deemed

too unwell by prison staff or (iii) posed a risk to the researchers. Consenting participants completed baseline and follow-up questionnaires.

The entirety of the intervention was delivered in a 30 min session. The session demonstrated use of the seven steps using the booklets and case studies developed in the focus groups. Prisoners were invited to attend subsequent follow-up sessions to assess progress and support their engagement with the intervention.

### Feasibility and acceptability measures

Data were collected on rates of recruitment, consent and retention for staff and prisoners. Reasons for non-participation and withdrawal were collected, where possible.

For outcome measures we assessed feasibility and acceptability by recording completion and follow-up rates. Typically, completion rates <50% are taken to indicate non-feasibility, >75% as indicating feasibility and 51%–74% as ambiguous—requiring modifications to design or implementation plans and reconsideration.

Our primary outcome proposed for a definitive trial was incidence of self-harm. Data on self-harm and/or attempted suicides were recorded at 3 months prior to baseline, baseline, postintervention and at 3 months follow-up (or up to point of release or transfer) from SystmOne using the search terms 'self-harm' and 'F213' (F213 is the title of the form used by the prison service to record incidents of self-harm behaviour). We explored recording of self-harm incidents through the prison ACCT register but found inconsistencies in the coding of data across the four sites.

Individual secondary outcomes at baseline and follow-up included measurement of quality of life using theEuroQol-visual analogue scale (EQ-VAS)[17] and depression using the Patient Health Questionnaire (PHQ-9).[18] The EQ-VAS is a self-rated questionnaire providing description of the subject's current health in five dimensions that is, mobility, self-care, usual activities, pain/discomfort and anxiety/depression and is rated into one of three degrees of disability (severe, moderate or none). The PHQ-9 is a well-validated tool for the measurement of depression with robust psychometric properties, reliability and validity in adult community populations.

Cost of usual care were estimated by: (i) completion of a self-report questionnaire reporting on access to the National Health Service (NHS) treatment before, during and after the study, (ii) staff interviews to ascertain the average time spent on each ACCT process and (iii) a case note review of 11 prisoner ACCT documents to record the amount of staff time involved in the ACCT procedure.

The costs of training included (i) the costs to release staff in attending the training sessions, (ii) the facilitator time in the delivery of the training and (iii) the number of training sessions, numbers of staff attending each session and the duration and timing of each training session. We obtained routinely collected electronic ACCT data consisting of individual and monthly ACCT information between January 2012 and December 2016. The time

period of the data collection was prescribed by the individual prison data collection protocols (online supplementary material, appendix A). We found that data were comparable from our four prison sites across this time period. Prior to 2012, the comparability of data and access to data were found to be limited and December 2016 was the latest date for which all prisons had complete data.

### Data analysis

Data were summarised, by prison, using descriptive summary statistics. The information included the description of the focus group participants, the number of training sessions and staff attending training sessions. The feasibility and success of recruitment of prisoners to the study is evaluated through summaries of the screening, eligibility, consent and recruitment processes.

A summary of the variability of available routine data across: outcomes, prison and wings (where available), and the estimated cost of usual care were informed using staff information and case review process. Delivery and implementation of the PST intervention were estimated using the number of training sessions, number of staff attending, standardised staff costs, facilitator time in the delivery of the session and preparation for each session alongside the cost of materials. Summary statistics for prisoner's baseline characteristics and outcomes for the incidence of self-harm behaviour, quality of life, depression and information on access to NHS treatment were recorded.

## RESULTS
### Feasibility assessment
#### Adapting and developing the materials

During 2015, staff and prisoners were nominated by each prison to participate in focus groups. Thirty-one staff participants attended (online supplementary material, appendix B). They comprised mainly operational 17/31 (55%) or managerial 6/31 (19%) staff with a mean age of 37 years. The majority were female 20/31 (66%), spoke English as their first language 27/31 (88%) and were British 27/31 (90%).

Six focus groups involving 67 prisoners, included mainly male prisoners 56/67 (83.6%) with a mean age of 39.8 years (SD 9.63). There were fewer prisoners on remand or first-time offenders involved in the focus groups, compared with recruited prisoners for the study (online supplementary material, appendix C). The process resulted in two gender-specific picture booklets and a series of exercises with associated case study scenarios that were used in the training and delivery of the intervention.

#### Coverage of staff training and recruitment

Two hundred eighty prison staff were trained between March 2015 and August 2016 (figure 1). Training was delivered by the research team to staff groups with a mean size of 8 staff (range of group sizes 2–19). Recruitment of

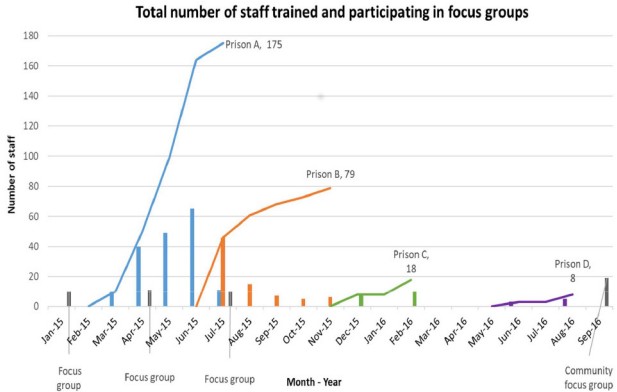

**Figure 1** Staff trained and participating focus groups.

staff to training sessions appeared to be acceptable and feasible.

Staff trained were mainly operational (120, 43%) or healthcare staff (78, 28%); other staff included a number of voluntary, managerial, admin, education and offender manager probation staff. Mean age of staff trained was 42 years, 59% were male and almost all spoke English as their first language and were British. Trained staff had spent a median of 8 years (range <1 month–36 years) working in the prison service (table 1).

### Screening and recruitment of prisoners

During the 3-month recruitment period at each site, a total of 281 prisoners were eligible to participate as per the study criteria. Of these, 106/281 (37%) were released or transferred to another prison site prior to invitation to attend an appointment in healthcare. The average time between identification of an eligible participant and meeting them to inform them about the studied varied between at each site between 1 and 3 weeks.

Of the remaining 175 (62%), 95/175 were not seen in healthcare for a variety of reasons. These included: 66/95 (69%) people who did not attend their appointment to be informed about the study following three consecutive invitations, 9/95 (9%) were considered too dangerous to approach, 6/95 (6%) lacked sufficient capacity, 5/95 (5%) were transferred or released prior to attending the appointment, 8/95 (8%) were not approached by the research team due to limited resources within the team and 1/95 person died (1%). Of the remaining 83 people, 6 (7%) attending the appointment were deemed not eligible reporting no incident of actual self-harm behaviour. For the remaining 75 people, 29/75 (39%) did not consent to take part leaving 48/75 (66%) consenting participants.

The median age of prisoners was 30 years (range 59–58 years). All but three were white British, and all spoke English as their first language. Two-thirds 32/48 (67%) were single and had never married; the majority smoked 39/48 (83%) and did not have a physical or learning disability (36/48 (77%) and 33/48 (69%)). Only a minority of prisoners recruited from prison B and none of those in prison D were on remand, while almost half

of prison A recruited prisoners were on remand 22/48 (46%). Only a quarter were first-time offenders 12/48 (25%), the number of times prisoners had been in prison ranged up to 50, with a median of 3 times. The median length of sentence was 27 months, with prisoners having spent a median of 3 months (range 2 days to 2 years) in their current and a median of 9 months left in prison (range 3 days to 15 years). For self-harm details, see online supplementary material, appendix D.

### Retention

Five out of 48 (10%) participants did not complete the intervention and withdrew from the study (figure 2), although general reasons were not provided for withdrawal. We tracked the transfer of 7/48 (15%) prisoners between our study sites. Transfer reasons included the progression of prisoners through their sentence (eg, from a local prison to our resettlement prison) or were unexpected due to a security breech.

### Adequacy of intervention delivery with prisoners who self-harmed

Between August 2015 and June 2016 delivery of the intervention by staff occurred for only two prisoners. At prison C, the research team had limited access to deliver the intervention and instead the prison decided to take the booklets and distribute them on the wings to target bullying. For the remaining 46/48 (96%) participants, the intervention was delivered by members of the research team in the healthcare unit.

The median time spent on intervention delivery was 40 min per prisoner (range 30–90 min). The overall time spent with the researcher, including the baseline assessment, intervention delivery, follow-up questionnaire for outcomes and qualitative interview averaged a median of 80 min (range 30 min up to 2 hours 30 min) over a period of one to seven contact appointments. In interviews, the intervention was acceptable to prisoners who received the intervention.[7]

### Acceptability of outcome measures
#### Use of routinely collected data to inform large-scale study

We found that reporting of self-harm data was complicated and recorded by several different methods, with variability in recording and differing definitions of self-harm across the four sites (online supplementary material, appendix E). Figure 3 shows the variability in monthly number of ACCTs opened at each site per 100 prisoners. The greatest variability of open ACCTs was displayed in prison C (our female site): online supplementary material, appendix F provide further details.

### Estimating the costs of usual care
#### Access to NHS services

All 48 prisoners had received some NHS service provision while in prison. Access to a general practitioner (range 1–10 appointments), pharmacist (daily drug dispensing) or duty nurse (range 1–35 appointments) appointments were the most cited points of contact; 35/48 (73%) prisoners reported accessing mental

**Table 1** Demographic information of staff trained

| | Prison A (n=175) | Prison B (n=79) | Prison C (n=18) | Prison D (n=8) | Total (n=280) |
|---|---|---|---|---|---|
| **Time working in the prison service (years)** | | | | | |
| N | 172 | 78 | 18 | 7 | 275 |
| Mean (SD) | 8.5 (8.93) | 13.0 (9.04) | 12.9 (8.45) | 12.1 (9.91) | 10.1 (9.16) |
| Median (range) | 6.0 (0.0–36.0) | 11.8 (0.1–35.3) | 12.5 (0.5–25.0) | 10.5 (1.3–29.2) | 8.0 (0.0–36.0) |
| **Time working in this prison (years)** | | | | | |
| N | 172 | 78 | 18 | 7 | 275 |
| Mean (SD) | 6.2 (7.48) | 11.2 (8.15) | 7.9 (7.61) | 9.0 (7.24) | 7.8 (7.96) |
| Median (range) | 3.3 (0.0–31.0) | 10.9 (0.1–35.3) | 4.5 (0.3–24.3) | 7.9 (1.1–20.8) | 5.5 (0.0–35.3) |
| **Since working here have you encountered an incident of self-harm?** | | | | | |
| Yes | 119 (68.0%) | 68 (86.1%) | 18 (100.0%) | 8 (100.0%) | 213 (76.1%) |
| No | 52 (29.7%) | 11 (13.9%) | 0 (0.0%) | 0 (0.0%) | 63 (22.5%) |
| Missing | 4 (2.3%) | 0 (0.0%) | 0 (0.0%) | 0 (0.0%) | 4 (1.4%) |
| **Most recent self-harm incident?** | | | | | |
| Within the past 7 days | 44 (37.0%) | 28 (41.2%) | 7 (38.9%) | 3 (37.5%) | 82 (38.5%) |
| Within the past month | 24 (20.2%) | 17 (25.0%) | 4 (22.2%) | 0 (0.0%) | 45 (21.1%) |
| Two months or more | 20 (16.8%) | 8 (11.8%) | 2 (11.1%) | 1 (12.5%) | 31 (14.6%) |
| Missing | 31 (26.1%) | 15 (22.1%) | 5 (27.8%) | 4 (50.0%) | 55 (25.8%) |
| **Type of incident?** | | | | | |
| Self-poisoning | 10 (8.4%) | 1 (1.5%) | 0 (0.0%) | 1 (12.5%) | 12 (5.6%) |
| Self-injury | 94 (79.0%) | 61 (89.7%) | 16 (88.9%) | 6 (75.0%) | 177 (83.1%) |
| Mixed self-poisoning and self-injury | 7 (5.9%) | 4 (5.9%) | 2 (11.1%) | 1 (12.5%) | 14 (6.6%) |
| Suicide | 7 (5.9%) | 2 (2.9%) | 0 (0.0%) | 0 (0.0%) | 9 (4.2%) |
| Missing | 1 (0.8%) | 0 (0.0%) | 0 (0.0%) | 0 (0.0%) | 1 (0.5%) |
| **Attended self-harm training?** | | | | | |
| Yes | 74 (42.3%) | 48 (60.8%) | 13 (72.2%) | 3 (37.5%) | 138 (49.3%) |
| No | 96 (54.9%) | 30 (38.0%) | 3 (16.7%) | 5 (62.5%) | 134 (47.9%) |
| Cannot recall | 4 (2.3%) | 1 (1.3%) | 2 (11.1%) | 0 (0.0%) | 7 (2.5%) |
| Missing | 1 (0.6%) | 0 (0.0%) | 0 (0.0%) | 0 (0.0%) | 1 (0.4%) |
| **Time since self-harm training?** | | | | | |
| N | 66 | 45 | 13 | 3 | 127 |
| Mean (SD) | 30.5 (38.22) | 20.4 (27.32) | 41.7 (53.84) | 19.0 (20.42) | 27.8 (36.64) |
| Median (range) | 12.0 (0.0–180.0) | 12.0 (1.0–120.0) | 12.0 (1.0–168.0) | 12.0 (3.0–42.0) | 12.0 (0.0–180.0) |

Continued

**Table 1** Continued

| | Prison A (n=175) | Prison B (n=79) | Prison C (n=18) | Prison D (n=8) | Total (n=280) |
|---|---|---|---|---|---|
| **Who provided this training?** | | | | | |
| Prison service | 59 (79.7%) | 42 (87.5%) | 8 (61.5%) | 3 (100.0%) | 112 (81.2%) |
| NHS | 3 (4.1%) | 0 (0.0%) | 2 (15.4%) | 0 (0.0%) | 5 (3.6%) |
| Nurse training | 2 (2.7%) | 1 (2.1%) | 0 (0.0%) | 0 (0.0%) | 3 (2.2%) |
| Other including University | 3 (4.1%) | 2 (4.2%) | 1 (7.7%) | 0 (0.0%) | 6 (4.3%) |
| Missing | 7 (9.5%) | 3 (6.3%) | 2 (15.4%) | 0 (0.0%) | 12 (8.7%) |
| **Length of training?** | | | | | |
| 1 hour | 19 (25.7%) | 0 (0.0%) | 1 (7.7%) | 0 (0.0%) | 20 (14.5%) |
| 2 hours | 9 (12.2%) | 4 (8.3%) | 1 (7.7%) | 1 (33.3%) | 15 (10.9%) |
| Half day | 16 (21.6%) | 14 (29.2%) | 1 (7.7%) | 1 (33.3%) | 32 (23.2%) |
| Full day | 13 (17.6%) | 18 (37.5%) | 7 (53.8%) | 1 (33.3%) | 39 (28.3%) |
| More than 1 day | 7 (9.5%) | 4 (8.3%) | 3 (23.1%) | 0 (0.0%) | 14 (10.1%) |
| Missing | 10 (13.5%) | 8 (16.7%) | 0 (0.0%) | 0 (0.0%) | 18 (13.0%) |

health services, 2 reported access to a psychological therapy. Just under a quarter, 11/48 (23%) had experienced a hospital admission and 13/48 (27%) (range 1–9) reported attending accident and emergency in the 3 months prior to incarceration. In all cases, admissions were related to synthetic cannabinoids intake, overdose, attempted hanging or feeling suicidal. There were no adverse events reported as part of the PST intervention during the study period.

We collected information from staff about how much time they spent conducting each element of the Assessment, Care in Custody and Teamwork (ACCT) process (online supplementary material, appendix G). Using an average time spent, each task in the ACCT process was assigned a proportionate salary costs (online supplementary material, appendix H).

We combined this staff information with data that we collected from the case review of 11 prisoners who had been on an ACCT during the study period. The 11 prisoners represented a total of 24 ACCTs documents that had been 'open' and 'shut' during their stay within the prison. For two prisoners, the ACCT was in use at the point of data collection providing a conservative estimate of cost. We added up the numbers of case reviews for each prisoner which ranged from 1 to 33, and added up the number of staff observations per ACCT document which ranged between 0 and 5520 staff observations. The total administrative costs for the 11 prisoners was estimated at £21 650, an average of £1968 per prisoner (range £375–£6416).

### Estimating the costs of training

Training costs included a notional hourly rate (of £15 per person) to cover the cost of releasing staff to attend the training session, and included the travel, preparation time and facilitator time in delivering the course and the cost of course materials. Across sites we estimated the training costs of between £500 and £6406 equating to a cost per prisoner of between £125 and £246 (online supplementary material, appendix H).

Overall, it proved feasible to gather resource information to provide a cost estimate of usual care, delivery of training and implementation of the intervention.

### Prisoner outcomes

Hundred per cent of those agreeing to participate in the study completed the baseline assessment. Follow-up times varied considerably, taking place a median of 2.8 months after recruitment but up to a maximum of 15 months for one prisoner (online supplementary material, appendix I). The timing of follow-up assessments fell into three clusters, the largest cluster taking place within the first 3 months post-recruitment, a further set taking place between 6 and 8 months postrecruitment in prison A. Follow-up was affected when access to prison A was halted for a 3-month period (figure 4). Overall, the average follow-up rate for questionnaire returns was 34/48 (71%) across the three sites. The changes in scores reflect them

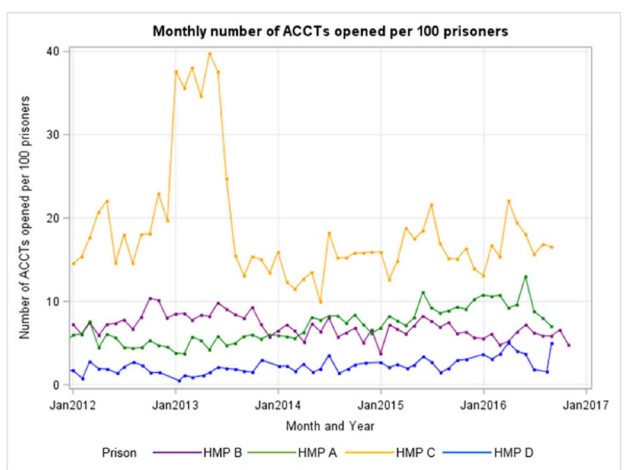

**Figure 2** Flow of participants through study. HMP, Her Majestys Prison.

as potentially useful outcome measures that could be used in a large-scale evaluation.

## Primary outcome: incidence of self-harm behaviour

Incidence of self-harm behaviour appeared to decrease over the life time of the project. At 3 months prior to

**Figure 3** Monthly numbers of ACCTs opened per 100 prisoners.

baseline, 32/48 (66%) prisoners had harmed themselves. This reduced to 9/48 (18%) prisoners at post-test.

## Secondary outcomes
### Quality of life
A total of 32/48 (66%) of individuals completed full information on the EQ-VAS. The baseline mean score (0.504, SD 0.34) fell postintervention (0.625, SD 0.347).

### Depression
At baseline, median scores were high at 18 and most prisoners had either moderately severe 18/48 (38%) or severe depression 20/48 (40%). Prisoners at follow-up had lower depression scores with just 7/48 (15%) classed as moderately severely depressed, and 13/48 (27%) as severely depressed (online supplementary material, appendix J).

## DISCUSSION
The study aimed to assess the feasibility and acceptability of adapting and implementing a brief PST intervention for prison staff and prisoners at risk of self-harm. Our results indicate that staff can be trained in using these skills, although most were unable to implement them

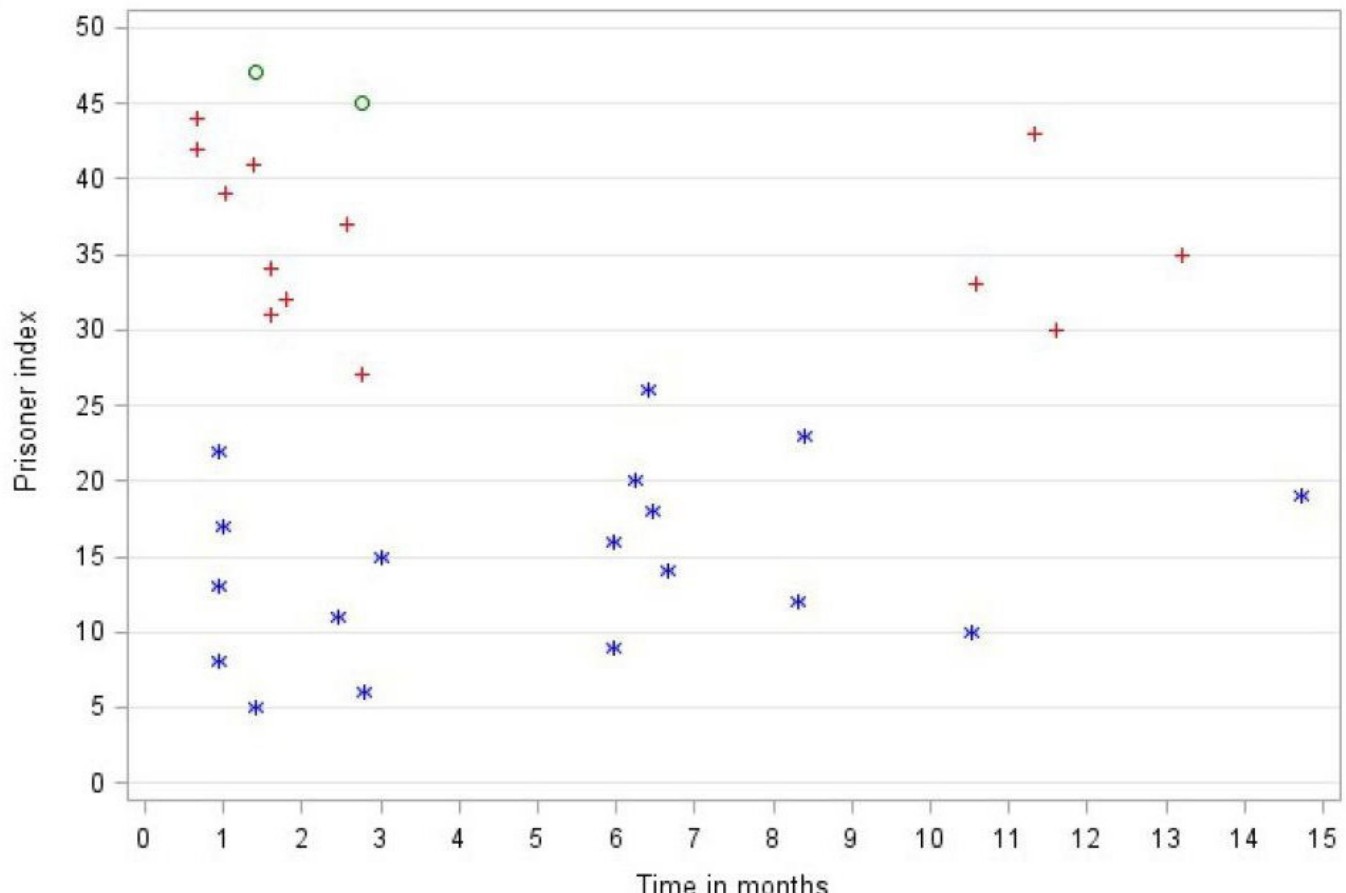

**Figure 4** Time between recruitment and questionnaire follow-up assessment. Key: * = HMP A, + = HMP B, o = HMP D.

with those who self-harmed. Prison staff faced severe time pressures, and limited resources making it difficult to accommodate the translation of knowledge into practice. This is a common problem in the design and implementation of complex interventions in organisations other than healthcare.[19] These findings emerged during the implementation phase. The brief nature of the training sessions themselves did not, perhaps, facilitate the expression of these doubts or tackle approaches to translation of skills into practice in a pressured environment.

Attrition from the study sample by prisoners was minimal due to the 30 min intervention design. Previous prison trials have demonstrated relatively high levels of attrition. In our study (despite a lack of access to one site for 3 months), we managed to produce encouraging follow-up rates (71%) suggesting that our outcomes were acceptable. Our findings are comparable with other pilot trials of self-harm in prisons[9 10] and trials of suicide prevention more broadly in the community.[20] We were able to track participants through our sites. This allowed us to collect follow-up data on seven participants who were released from prison A, and either returned back to the same prison during the study period or were transferred to prison C prior to release. Prison function is therefore an important consideration. Turnover of prisoners at our local prison sites (eg, prisons A and B) was considerably greater than in our resettlement prison. This finding is

supported elsewhere with data provided from prison A in a recent Inspectorate report showing that 430/1109 (38%) were imprisoned for <3 months in 2017. Prisoners followed from prison A through to prison C were notably in a better position to engage with training when in the resettlement prison. This system of 'tracking' participants provides a potential mechanism to ensure adequate follow-up in a large-scale study.

There were limitations with the development of our economic protocol in the assumptions made with regard to costs for usual care which are not necessarily representative. Access, quality and consistency of these data varied across the prison sites and led us to conclude that such routine data could only be used to measure the impact of any future evaluation if additional data were provided or stricter collection protocols and monitoring were deployed. We also propose that any new study should include individual self-report information and information from local and national data sources. This method is not dissimilar to other data collection mechanisms in two pilot trials of self-harm in UK prisons where prisoners report suicidal behaviours, thoughts and feelings.[9 21]

In designing a large-scale study, we have sufficient information to inform our outcomes of measurement and feasibility of data collection. However, alternative implementation mechanisms need to be identified prior to any large-scale study. Our qualitative findings[11] suggest

two alternative options: first, use of trusted prisoners as 'problem support mentors' to deliver the skills to peers on the wings and/or second, delivery of problem-solving skills to prisoners through education classes.

## CONCLUSIONS

The study suggests that the modified version of PST, adapted for training, was acceptable to prisoners. Although the study demonstrated that it was currently not feasible to deliver the intervention using prison staff it provides insight into how such an intervention with prisoner-staff involvement can be adapted for use in a different environment.

**Author affiliations**
[1]Health Sciences, Unviersity of York, York, UK
[2]Faculty of Medicine and Health, University of Leeds, Leeds, UK
[3]Academic Unit of Psychiatry, University of Leeds, Leeds, UK
[4]Clinical Trials Research Unit, University of Leeds, Leeds, UK
[5]Sociology and Social Policy, University of Leeds, Leeds, UK
[6]Centre for Health Economics, University of York, York, UK
[7]Spectrum Community Health CIC, Wakefield, UK

**Acknowledgements** The authors would like to thank Spectrum Community Health CIC and previously Leeds Community HealthCare Trust who agreed to host this study. The authors would like to thank Christine Butt who helped with the collection of data at one prison site. The authors would also like to acknowledge the contributions of all the prison staff, prisoners, for support from the North East and Yorkshire Area Safer Custody Team and prison Governors for their willingness to take part and support this study.

**Contributors** AP, AH and MGW designed and conducted most of the study with considerable input from AKH. AWH and AF took the lead in performing the statistical analyses together and JG was the lead for analysing the qualitative interviews with AP. GR and NW led the development of the economic protocol and information on the study costs. NW supported access to the prison sites and all authors provided input into the writing of the manuscript.

**Funding** This paper presents independent research funded by the NIHR under its Research for Patient Benefit (RfPB) Programme (Grant Reference Number PB-PG-0211-24122).

**Competing interests** None declared.

**Patient consent for publication** Not required.

**Ethics approval** Ethical approval for the study was obtained for phase I from NHS REC approval (NRES, North East York, 28 October 2014) and NOMS (1 September 2014) and phases II–V (Bristol REC Centre, London South East, 6 January 2015) from NHS REC approval, NOMS (6 March 2015) and the Department of Health Sciences at the University of York for all phases (11 December 2014). As the material was adapted and developed for appropriate use within each prison, we were granted one substantive amendment to the project from all parties during July 2015.

**Provenance and peer review** Not commissioned; externally peer reviewed.

**Data availability statement** Data are available on reasonable request.

**ORCID iDs**
Amanda Perry http://orcid.org/0000-0002-0279-1884
Allan House http://orcid.org/0000-0001-8721-8026

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
