## [Reviewer comments · BMJ Open]

ARTICLE DETAILS

TITLE (PROVISIONAL)	Problem-Solving Training: assessing the feasibility and acceptability of delivering and evaluating a problem-solving training model for front-line prison staff and prisoners who self-harm.
AUTHORS	perry, Amanda; Waterman, Mitchell; House, Allan; Wright-Hughes, Alexandra; Greenhalgh, Joanne; Farrin, Amanda; Richardson, Gerry; Hopton, Ann; Wright, Nat

VERSION 1 – REVIEW

REVIEWER	Professor Eddie Kane University of Nottingham UK
REVIEW RETURNED	20-Nov-2018

GENERAL COMMENTS	Interesting paper in an important area. If a way could be found to effectively deliver the training to prisoners given that staff appear not to be able to do so (usual reasons) it would be worth a scaled up trial with the limitations etc you highlight being addressed in the protocol. The results from a small sample are encouraging.
---

REVIEWER	Dr Kerry Gutridge Research Associate and Trial Manager Centre for Women's Mental Health Division of Psychology and Mental Health School of Health Sciences Faculty of Biology, Health and Medicine University of Manchester Oxford Road Manchester M13 9PL United Kingdom
REVIEW RETURNED	07-Dec-2018

GENERAL COMMENTS	I would like a bit more detail regarding the content of the problem solving therapy so readers unfamiliar with the approach have a clearer idea of its key components. I would also like to see more information on how the focus groups refined the training. It would be helpful if the authors explained the rationale for only using SystemOne to collect self-harm incidents. Did they consider using prison databases and self report, in addition to healthcare records? I think that the description of the costs that were collected during the trial could be clearer. Was an attempt made to cost access to NHS services in prison as well as the costs of ACCT and the training?
--

VERSION 1 – AUTHOR RESPONSE

In response the following comments:

1. I would like a bit more detail regarding the content of the problem solving therapy so readers unfamiliar with the approach have a clearer idea of its key components.

I have added in an extra paragraph in the 'introduction to the paper' which also puts problem-solving into context with the World Health Organisation and its publication in 2016. I have edited the paragraph describing the intervention details to add further comment

2. I would also like to see more information on how the focus groups refined the training.

I have added a couple of extract sentences to explain how the focus groups helped to refine the training materials.

3. It would be helpful if the authors explained the rationale for only using SystemOne to collect self-harm incidents.

Did they consider using prison databases and self report, in addition to healthcare records?

We did use the prison database information in addition to the SystemOne, but we found inconsistencies about how self-harm was recorded. I have added a sentence to reflect this.

3. I think that the description of the costs that were collected during the trial could be clearer. Was an attempt made to cost access to NHS services in prison as well as the costs of ACCT and the training?

I agree with these comments I have gone back to this section of the paper and edited the text around the methodology of the costs and also the results. I have added in some sub-headings to provide a division between the costs of usual care and those of training and intervention delivery. I have added in a paragraph which gives the self-report results of the prisoners in their use of health care before during and after the study. This was previously missing.